# Impact of Coronavirus Disease 2019 on Out-of-Hospital Cardiac Arrest Survival Rate: A Systematic Review with Meta-Analysis

**DOI:** 10.3390/jcm10061209

**Published:** 2021-03-15

**Authors:** Magdalena J. Borkowska, Miłosz J. Jaguszewski, Mariusz Koda, Aleksandra Gasecka, Agnieszka Szarpak, Natasza Gilis-Malinowska, Kamil Safiejko, Lukasz Szarpak, Krzysztof J. Filipiak, Jacek Smereka

**Affiliations:** 1Department of Research Outcomes, Maria Sklodowska-Curie Białystok Oncology Centre, 15-027 Białystok, Poland; ptmk.kontakt@gmail.com (M.J.B.); kodamar2@gmail.com (M.K.); ksafiejko@onkologia.bialystok.p (K.S.); l.szarpak@fmcgroup.org (L.S.); 21st Department of Cardiology, Medical University of Gdansk, 80-952 Gdansk, Poland; mjaguszewski@gumed.edu.pl (M.J.J.); tasza.gilis@gmail.com (N.G.-M.); 3Department of General Pathomorphology, Medical University of Bialystok, 15-089 Bialystok, Poland; krzysztof.filipiak@wum.edu.pl; 41st Chair and Department of Cardiology, Medical University of Warsaw, 02-097 Warsaw, Poland; 5Department of Research Outcomes, Maria Sklodowska-Curie Medical Academy in Warsaw, 03-411 Warsaw, Poland; agnieszkaszarpak2019@gmail.com; 6Department of Research Outcomes, Polish Society of Disaster Medicine, 05-090 Raszyn, Poland; jacek.smereka@umed.wroc.pl; 7Department of Emergency Medical Service, Wroclaw Medical University, 50-367 Wroclaw, Poland

**Keywords:** out-of-hospital cardiac arrest, cardiopulmonary resuscitation, COVID-19, SARS-CoV-2, outcome, survival rate, systematic review, meta-analysis

## Abstract

Out-of-hospital cardiac arrest (OHCA) is a challenge for medical staff, especially in the COVID-19 period. The COVID-19 disease caused by the SARS-CoV-2 coronavirus is highly infectious, thus requiring additional measures during cardiopulmonary resuscitation (CPR). Since CPR is a highly aerosol-generating procedure, it carries a substantial risk of viral transmission. We hypothesized that patients with diagnosed or suspected COVID-19 might have worse outcomes following OHCA outcomes compared to non-COVID-19 patients. To raise awareness of this potential problem, we performed a systematic review and meta-analysis of studies that reported OHCA in the pandemic period, comparing COVID-19 suspected or diagnosed patients vs. COVID-19 not suspected or diagnosed group. The primary outcome was survival to hospital discharge (SHD). Secondary outcomes were the return of spontaneous circulation (ROSC), survival to hospital admission or survival with favorable neurological outcomes. Data including 4210 patients included in five studies were analyzed. SHD in COVID-19 and non-COVID-19 patients were 0.5% and 2.6%, respectively (odds ratio, OR = 0.25; 95% confidence interval, CI: 0.12, 0.53; *p* < 0.001). Bystander CPR rate was comparable in the COVID-19 vs. not COVID-19 group (OR = 0.88; 95% CI: 0.63, 1.22; *p* = 0.43). Shockable rhythms were observed in 5.7% in COVID-19 patients compared with 37.4% in the non-COVID-19 group (OR = 0.19; 95% CI: 0.04, 0.96; *p* = 0.04; I^2^ = 95%). ROSC in the COVID-19 and non-COVID-19 patients were 13.3% vs. 26.5%, respectively (OR = 0.67; 95% CI: 0.55, 0.81; *p* < 0.001). SHD with favorable neurological outcome was observed in 0% in COVID-19 vs. 3.1% in non-COVID-19 patients (OR = 1.35; 95% CI: 0.07, 26.19; *p* = 0.84). Our meta-analysis suggests that suspected or diagnosed COVID-19 reduces the SHD rate after OHCA, which seems to be due to the lower rate of shockable rhythms in COVID-19 patients, but not due to reluctance to bystander CPR. Future trials are needed to confirm these preliminary results and determine the optimal procedures to increase survival after OHCA in COVID-19 patients.

## 1. Introduction

Sudden cardiac arrest (SCA) is a challenge for medical staff in both the prehospital and hospital settings and requires immediate cardiopulmonary resuscitation (CPR) [1]. Nearly 300,000 cases of out-of-hospital cardiac arrest (OHCA) are recorded annually in North America, and this number is close to 250,000 in Europe [2,3,4,5]. The OHCA survival rate ranges from 8% to 12%. However, this range applies to patients in whom SCA occurred in the presence of emergency medical service (EMS) teams, facilitating resuscitation measures immediately after the diagnosis of OHCA [6].

For almost a year, the world has been struggling with the pandemic of the new coronavirus SARS-CoV-2, which causes the highly contagious disease COVID-19 [7]. The drip route, bioaerosols and direct contact with contaminated surfaces are the most common routes of SARS-CoV-2 transmission [8]. As of 19 February 2021, 110,349,428 cases of COVID-19 infection have been reported worldwide, with a mortality rate of 2.2% [9]. There is accumulating evidence that SARS-CoV-2 infection leads to endothelial damage and causes microvascular thrombosis, which increases the risk of SCA and impairs patient prognosis [10,11,12]. SARS-CoV-2 may also indirectly affect SCA outcomes by altering the capacity of the community and EMS teams to respond to OHCA, as exemplified by numerous studies indicating extended EMS travel time in the COVID-19 period compared to the pre-COVID-19 period [13,14]. We hypothesized that the reluctance to bystander CPR deteriorates OHCA outcomes in the COVID-19 patients. To raise awareness of this potential problem, we performed a systematic review and meta-analysis of studies that reported OHCA in the pandemic period, comparing COVID-19 suspected or diagnosed patients vs. COVID-19 not suspected or diagnosed group.

## 2. Materials and Methods

This systematic review and meta-analysis was performed following Preferred Reporting Items for Systematic reviews and Meta-analysis (PRISMA) guidelines [15] and with Meta-analysis of Observational Studies in Epidemiology (MOOSE) statement [16].

### 2.1. Search Strategy

Two investigators (M.J.B. and L.S.) independently searched for published clinical trials or observational studies indexed in MEDLINE, EMBASE, Web of Science, Scopus and the Cochrane databases from inception to 19 February 2021 using the following terms: “out-of-hospital cardiac arrest” OR “OHCA” OR “cardiac arrest” AND “survival” OR “mortality” AND “SARS-CoV-2” OR “COVID-19”. We limited the search to English-language studies. A manual search for additional pertinent studies and review articles using references from the retrieved articles was also completed.

### 2.2. Inclusion Criteria and Exclusion Criteria

The PECOS strategy, consisting of patient, exposure, comparison and outcome, was used as a tool to ensure focused clinical questions [17]. The prespecified criteria for studies included in the meta-analysis were original papers of randomized controlled trials (RCTs) or cohort studies of (P) participants, adult patients with out-of-hospital cardiac arrest; (E) COVID-19 suspected or diagnosed patients; (C) COVID-19 not suspected or diagnosed patients; (O) outcomes, detailed information for survival, return of spontaneous circulation and length of hospital stay; (S) study design, observational studies comparing resuscitation effects in patients with cardiac arrest. Studies were excluded if they were reviews, case reports, conference or poster abstracts or articles not containing original data or comparator group.

### 2.3. Data Extraction

The titles and abstracts were screened for relevance by 2 authors (L.S. and J.S.) independently; if differences were found, they were discussed until a consensus was reached. The manuscripts of selected titles/abstracts were assessed for inclusion, and the authors were contacted if further information was required. Using the selection criteria enlisted above, the three reviewers (M.J.B., L.S. and J.S.) independently identified the papers to be included and excluded, and data from the included papers were extracted using predefined extraction flow sheets. The following information was extracted: authors, year of publication, study design, COVID-19 characteristics, number of participants, mean age of participants, prehospital resuscitation characteristics (Table 1).

The primary endpoint was survival to hospital discharge (SHD). Secondary outcomes included advanced life support (ALS) implementation in prehospital care, prehospital ROSC, and survival to hospital admission or survival with favorable neurologic.

### 2.4. Quality Assessment

Two reviewers (M.J.B. and L.S.) independently extracted individual study data and evaluated studies for risk of bias using a previously piloted standardized form and the Newcastle–Ottawa scale [18]. The three major domains of quality of a study covered by this tool were the selection of participants, comparability of cohorts and outcome assessment. Any disagreement was resolved by discussion with a third investigator (J.S.).

### 2.5. Statistical Analysis

Statistical analysis was performed with Review Manager (RevMan) software, version 5.4 (Cochrane Collaboration, Oxford, UK). The Mantel-Haenszel method was used to analyze dichotomous outcomes, and results were reported as odds ratio (ORs). For continuous measures (procedure time), we calculated the mean differences (MD). A random-effect model was applied to analyze the data. Results are presented as odds ratios (ORs) with 95% confidence intervals (CI) for dichotomous measures. When the continuous outcome was reported in a study as median, range and interquartile range, we estimated means and standard deviations using the formula described by Hozo et al. [19]. We quantified heterogeneity in each analysis by the tau-squared and I-squared statistics. Heterogeneity was detected with the chi-squared test with n − 1 degree of freedom, which was expressed as I^2^. Values of I^2^ > 50% and >75% were considered to indicate moderate and significant heterogeneity among studies, respectively. A *p*-value less than 0.05 was judged to be statistically significant.

### 2.6. Role of the Funding Source

This study was not supported by any funding source. The corresponding author had full access to all the data and had the final decision to submit it for publication.

## 3. Results

### 3.1. Study Characteristics

According to the search strategy, a total of 242 related studies were retrieved. After removing duplicate studies and excluding irrelevant titles or articles, 29 articles remained. After detailed examination, five studies (4210 patients) were included in the final analysis [20,21,22,23,24]. The flow chart summarizing the process of study selection is shown in Figure 1. All studies were retrospective studies. Of the five studies, there was one from each of the following countries: France [20], Italy [21], Republic of Korea [22], UK [23] and Sweden [24]. Three studies were based on national registries [20,22,24] and two were regional studies conducted in the Province of Lombardia [21] and in London [23]. Three studies were published in 2020 [20,21,22] and two in 2021 [23,24].

The principal features of the included studies are displayed in Table 1. Detailed characteristics of the studies included in the meta-analysis are presented in the Appendix A. All studies included patients with OHCA of medical origin, attended by EMS. The mean age of patients included in the studies ranged from 67 ± 18 to 77 ± 2.3 years (Appendix A). The majority of patients included in the studies were male, except for the Korean study, where 60% of patients suffering from OHCA in the COVID-19 period were women (Appendix A). There were substantial differences between the studies regarding the percentage of witnessed OHCA, from 64% in the French study to 9–10% in the Italian study. Moreover, the percentage of bystander CPR differed and ranged from over 60% in all patients in the UK study to only 10% in the COVID-19 patients in the Korean and Italian study. In the non-COVID-19 group, the shockable initial rhythm was observed in 22.8% in the Sweden study, 20% in the UK study and about 10% in other studies. On the contrary, shockable rhythm was observed in 0-10% of COVID-19 patients only, depending on the study. The mean time from call to arrival differed from about 10 min in the UK study to about 20–25 min in the French and Korean studies. Regardless of the differences in geographical setting and baseline characteristics, there were no significant differences in the age, gender ratio, witnessed OHCA or bystander CPR in COVID-19 vs. non-COVID-19 group (Appendix A). All studies consistently found lower survival to discharge rate in COVID-19 patients, compared to non-COVID-19 patients (Table 1).

### 3.2. OHCA Outcomes

Five studies reported survival to hospital discharge [20,21,22,23,24]. SHD in the COVID-19 group was 0.5% and was significantly lower than in the non-COVID-19 group (2.6%; OR = 0.25; 95% CI: 0.12, 0.55; *p* < 0.001; I^2^ = 0%; Figure 2).

ROSC was reported in five studies and was 13.3% for COVID-19 group compared to 26.5% for non-COVID-19 group (OR = 0.67; 95% CI: 0.55, 0.81; *p* < 0.001; I^2^ = 20%).

SHD with favorable neurological was reported in one study [22] and amounted to 0% in the COVID-19 patients vs. 3.1% in the non-COVID-19 group (OR = 1.35; 95% CI: 0.07, 26.19; *p* = 0.84). A detailed summary of the survival outcomes is presented in Table 2.

### 3.3. Cardiopulmonary Resuscitation Parameters

Cardiac arrest was witnessed by bystanders in 52.6% in COVID-19 group compared to 54.9% in not COVID-19 group (OR = 0.99; 95% CI: 0.84, 1.17; *p* = 0.93; I^2^ = 5%; Appendix A). Bystander CPR application was reported in five studies and obtained 51.4% vs. 49.1%, respectively, for COVID-19 and non-COVID-19 patients (OR = 0.88; 95% CI: 0.63, 1.22; *p* = 0.43; I^2^ = 60%; Appendix A).

ALS implementation towards patients with OHCA in with and without suspicion or diagnosis was reported in two articles and amounted to 49.7% vs. 55.8% respectively (OR = 0.87; 95% CI: 0.66, 1.13; *p* = 0.29; I^2^ = 0%; Appendix A).

Shockable rhythms were observed in 5.7% in COVID-19 patients compared with 37.4% in non-COVID-19 group (OR = 0.19; 95% CI: 0.04, 0.96; *p* = 0.04; I^2^ = 95%; Appendix A).

In turn, studies showing the frequency of use mechanical chest compression devices in prehospital care varied and amounted to 6.7% for COVID-19 patients compared with 25.5% for non-COVID-19 patients (OR = 0.72; 95% CI: 0.25, 2.09; *p* = 0.54; I^2^ = 0%; Appendix A).

## 4. Discussion

In this meta-analysis of five studies including 4210 adult patients with OHCA in the COVID-19 period, we found that the rates of ROSC and SHD in suspected or confirmed COVID-19 patients were significantly lower than in the non-COVID-19 patients (0.6% vs. 3.8%). In addition, COVID-19 had a substantially lower rate of initial shockable rhythms compared to non-COVID-19 patients (37.4% vs. 5.7%). On the contrary, the rate of bystander CPR and ALS implementation was comparable in both groups.

The increase in mortality might hypothetically be for two reasons. First, COVID-19 has changed the risk–benefit balance for CPR. Since CPR is a highly aerosol-generating procedure, it carries a substantial risk of viral transmission [25]. Hence, it should be done with full personal protective equipment (PPE) [26,27,28], which remains challenging for clinicians. To support this, several studies showed the negative influence of PPE on rescuer comfort and ability to carry out CPR [29,30,31]. In addition, it could be hypothesized that the risk of viral transmission might discourage bystander CPR. However, our data do not support this hypothesis, since the rate of CPR was comparable in COVID-19 and non-COVID-19 patients.

Second, the increase in mortality might relate to a comparison of two different diseases. In non-COVID-19 patients, OHCAs are mainly caused by ischemic heart disease, while in COVID-19 patients, additional factors, including hypoxia, myocarditis, pulmonary embolism, microvascular thrombosis and endothelial dysfunction play a role [32,33,34]. This explanation is supported by the substantial difference in the frequency of shockable rhythms between COVID-19 and non-COVID-19 patients [35,36]. Noteworthy, SARS-CoV-2 infection increased the incidence of OHCA [14,37]. Concurrently, the severe course of COVID-19 infection leading to cardiac arrest probably has high mortality in itself [38,39,40,41], indicating that patients with ROSC have to survive both the usual SCA complications and the COVID-19 disease [42,43]. Based on our data, the reduced SHD of patients with COVID-19 following OHCA seems to result rather from the COVID-19 disease than from the worse CPR quality delivered by bystanders or EMS.

## 5. Strengths and Limitations

Our analysis has several limitations. First, it is based on only five studies that differed regarding geographical setting and baseline characteristics, which might have been a source of bias. Second, we were unable to access individual patient data, so the potential influences of comorbidities (age, diabetes, length of COVID-19 symptoms), cardiovascular medications and other parameters may be underestimated. Third, based on the design of studies included in this meta-analysis, we had to combine the suspected and confirmed patients of COVID-19, which limits the ability of our study to truly compare outcomes between COVID-19 vs. non-COVID-19 patients. Finally, we could not include in our analysis data on no flow- and low flow-time, which are important prognostic parameters in OHCA, because these times were not available in the source articles.

The study also has strengths. We performed a systematic review according to Cochrane’s methodology. To the best of our knowledge, this was the first meta-analysis focused on OHCA in the pandemic period comparing COVID-19 suspected or diagnosed patients to non-COVID-19 patients. Previous studies analyzed the first months of the pandemic with the corresponding periods in the years preceding the pandemic [38,39,40,41], making them more prone to bias. In contrast, this meta-analysis was based on the analysis of the incidence of OHCA during the COVID-19 pandemic, increasing the reliability of the results.

## 6. Conclusions

In conclusion, available evidence from the present meta-analysis suggests that the co-existing suspected or confirmed COVID-19 in the case of OHCA reduces SHD and ROSC, which is likely due to the initially more severe state of the suspected or confirmed COVID-19 patients, as indicated by the lower rate of shockable rhythms, and not due to the lower frequency of bystander CPR and ALS implementation.

## Figures and Tables

**Figure 1 jcm-10-01209-f001:**
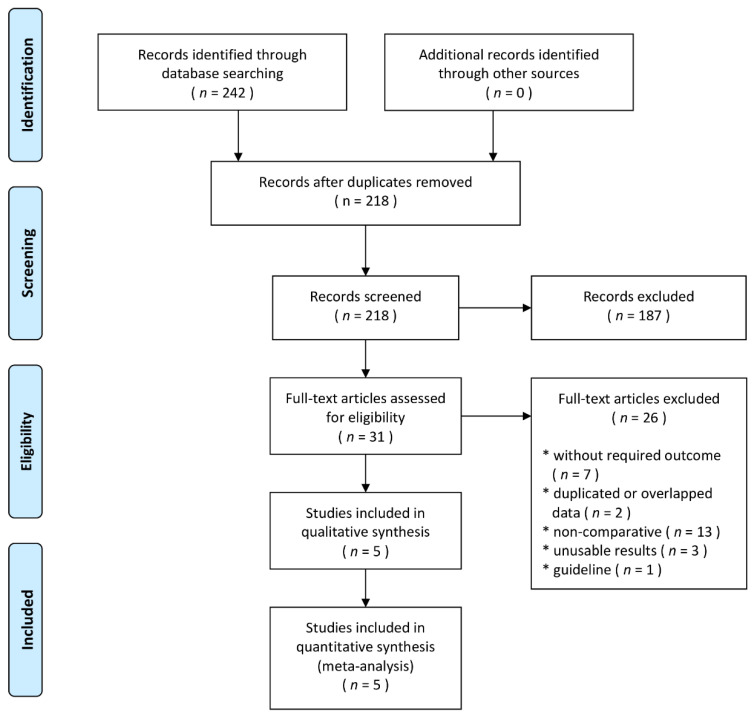
Flow diagram showing stages of database searching and study selection as per Preferred Reporting Items for Systematic reviews and Meta-analysis (PRISMA) guideline.

**Figure 2 jcm-10-01209-f002:**
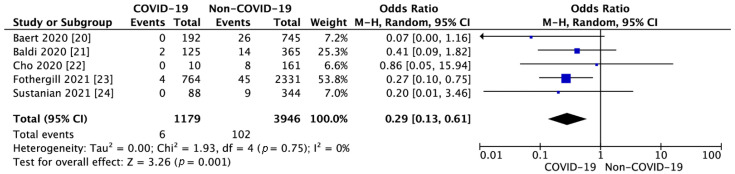
Forest plot of survival to hospital discharge in COVID-19 and non-COVID-19 group. The center of each square represents the weighted odds ratios for individual trials, and the corresponding horizontal line stands for a 95% confidence interval. The diamonds represent pooled results.

**Table 1 jcm-10-01209-t001:** Characteristics between COVID-19 and non-COVID-19 patients in the included studies.

Study	Country	Study Design	COVID-19 Status	Number of Patients	Age, Years (Mean ± SD)	Sex (Male), No./Total (%)	Bystander Witnessed No./Total (%)	Bystander CPR No./Total (%)	Shockable Initial Rhythm No./Total (%)	Time from Call to Arrival (Mean ± SD)	Survival to Discharge No./Total (%)	NOS Score
Baert et al. 2020 [20]	France	Multi-centre retrospective study	COVID-19	197	67 ± 18	117/197 (59.4)	126/197 (64.0)	99/197 (50.3)	8/197 (4.1)	25 ± 22	0/192 (0.0)	Fair
			Non-COVID-19	808	69 ± 16	559/808 (69.2)	522/808 (64.6)	401/808 (49.6)	79/806 (9.8)	23 ± 17	26/745 (3.5)	
Baldi et al. 2020 [21]	Italy	Single-centre retrospective study	COVID-19	125	77 ± 2.3	83/125 (66.4)	25/125 (20)	13/125 (9.6)	8/125 (9.1)	16.6 ± 1.7	2/125 (2.3)	Fair
			Non-COVID-19	365	76.8 ± 2.8	238/365 (65.2)	33/365 (9.0)	76/365 (39.2)	28/365 (12.4)	14.5 ± 1.3	14/365 (6.2)	
Cho et al. 2020 [22]	Republic of Korea	Multi-centre retrospective study	COVID-19	10	73.3 ± 4.3	4/10 (40.0)	10/10 (100)	1/10 (10.0)	0/10 (0)	24.5 ± 4.6	0/10 (0)	Good
			Non-COVID-19	161	72.3 ± 3.2	104/161 (64.6)	120/161 (74.5)	57/161 (35.4)	15/161 (9.3)	19.5 ± 1.7	8/161 (5.0)	
Fothergill et al. 2021 [23]	UK	Single-centre retrospective study	COVID-19	766	70 ± 18	468 (61.2)	216/393 (55.0)	257/393 (65.4)	24/393 (6.2)	11 ± 1.8	4/764 (0.5)	Good
			Non-COVID-19	2356	71 ± 19	1371 (58.3)	390/742 (52.6)	461/742 (62.1)	144/742 (19.5)	9.7 ± 1.3	45/2331 (1.9)	
Sultanian et al. 2021 [24]	Sweden	Observational registry-based study	COVID-19	88	66.5 ± 18.4	59 (67.0)	37 (42.0)	48 (54.5)	6 (6.8)	11.8 ± 2.2	0 (0.0)	Good
			Non-COVID-19	334	70.6 ± 16.4	241 (72.2)	158 (47.3)	188 (56.3)	63 (18.9)	13 ± 2.3	9 (2.7)	

Legend: CPR = Cardiopulmonary resuscitation; NS = Not specified; OHCA = out-of-hospital cardiac arrest; SD = Standard deviation; NOS = Newcastle-Ottawa Scale.

**Table 2 jcm-10-01209-t002:** Survival outcomes in included studies.

Parameter	No. of Studies	Cases in COVID-19 Suspected or Diagnosed Group	Cases in COVID-19 Not Suspected or Diagnosed Group	OR (95% CI)	*p* Value	I^2^ Statistics
Death in the field	2	401/518 (77.4%)	650/1107 (58.7%)	2.02 (0.83, 4.92)	0.001	92%
Transport with ongoing CPR	1	11/125 (8.8%)	23/365 (6.3%)	1.43 (0.68, 3.03)	0.34	NA
ROSC	5	108/812 (13.3%)	637/2405 (26.5%)	0.67 (0.55, 0.81)	<0.001	20%
Survival to hospital admission	3	41/528 (7.8%)	207/1268 (16.3%)	0.54 (0.19, 1.52)	0.008	70%
Favourable neurological outcome at discharge	1	0/10 (0%)	5/161 (3.1%)	1.35 (0.07, 26.19)	0.84	NA

Legend: CI = Confidence interval; CPR = Cardiopulmonary resuscitation; OR = Odds ratio; ROSC = Return of spontaneous circulation.

## Data Availability

Not applicable.

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
