# Peer review of "Impact of Coronavirus Disease 2019 on Out-of-Hospital Cardiac Arrest Survival Rate: A Systematic Review with Meta-Analysis"

_jcm, 2021, doi:10.3390/jcm10061209_

Round 1

Reviewer 1 Report

Review

The authors performed a meta-analysis on out-of-hospital cardiac arrest survival rates in Covid-19 and Non-Covid-19 patients. After screening 217 records, they included 4 studies with approximately 3’800 patients. Key finding was a lower survival to hospital discharge in Covid-19 patients (OR 0.25).

Major comments:

The introduction and discussion are lengthy and difficult to read. Please focus and shorten the manuscript  

  1. For me as a reader, it is not clear enough why you performed the meta-analysis. I suggest to start the introduction with a brief statement on what we know on SHD, ROSC, and other important outcomes from the pre-Covid area (not more than 2 senteces): Then elaborate on what has changed with Covid-19 and what could have changed. From there, justify the need for a meta-analysis.
  2. Again, lengthy and difficult to read. I suggest that after you presented the key findings in the first paragraph, you focus on the two most likely explanations for your findings:
    1. The increase in mortality might relate to rescuers, eg reluctance and/or PPE
    2. The increase in mortality might relate to a comparison of two different diseases. In Non-Covid patients, OHCAs are mainly caused by ischaemic heart disease while in Covid patients, additional factors may play an important role, eg hypoxic arrests, myocarditis, pulmonary embolism, endothelial dysfunction. The difference in shockable rythms would support the notion of two different conditions. In addition, a Covid infection leading to a cardiac arrest probably has a high mortality in itself, meaning that patients surviving the initial resuscitation not only have to survive all usual complications but also their Covid disease.
  3. Shockable rhythm. I was surprised to see that the large difference in shockable rhythm did not result in a significant statistical difference (page 3, line 82). Doing a simple chi-square test with data extracted from your table 1 yielded a highly significant difference (see print screen below). What is the explanation for this discrepancy?

Category 1

Category 2

Marginal Row Totals

Group 1

40   (79.15)   [19.36]

685   (645.85)   [2.37]

725

Group 2

266   (226.85)   [6.76]

1812   (1851.15)   [0.83]

2078

Marginal Column Totals

306

2497

2803    (Grand Total)

The chi-square statistic is 29.3191. The p-value is < 0.00001. Significant at p < .05.

Minor comments

  1. Should be more focused and should include a “justification” of doing this meta-analysis
  2. Page 2, line 49. Please provide a reference for the 7%
  3. Page 2, line 61. Please provide a reference for the 2.1%
  4. Page 2, line 84. Please provide a reference for the PECOS procedure
  5. Page 3, table 2. Do the studies included in your meta-analysis indicate, which proportion of resuscitations were performed with rescuers wearing full PPE? If yes, please include this information
  6. Page 5, line 165 “lower” instead of “rower”
  7. Page 5, line 181. What does “consubstantial Covid-19” mean?

Author Response

Reviewer 1

The authors performed a meta-analysis on out-of-hospital cardiac arrest survival rates in Covid-19 and Non-Covid-19 patients. After screening 217 records, they included 4 studies with approximately 3’800 patients. Key finding was a lower survival to hospital discharge in Covid-19 patients (OR 0.25).

Major comments:

The introduction and discussion are lengthy and difficult to read. Please focus and shorten the manuscript 

  1. For me as a reader, it is not clear enough why you performed the meta-analysis. I suggest to start the introduction with a brief statement on what we know on SHD, ROSC, and other important outcomes from the pre-Covid area (not more than 2 senteces): Then elaborate on what has changed with Covid-19 and what could have changed. From there, justify the need for a meta-analysis.

We thank the Reviewer for this comment. We shortened the introduction and re-wrote the discussion.

  1. Again, lengthy and difficult to read. I suggest that after you presented the key findings in the first paragraph, you focus on the two most likely explanations for your findings:
    1. The increase in mortality might relate to rescuers, eg reluctance and/or PPE
    2. The increase in mortality might relate to a comparison of two different diseases. In Non-Covid patients, OHCAs are mainly caused by ischaemic heart disease while in Covid patients, additional factors may play an important role, eg hypoxic arrests, myocarditis, pulmonary embolism, endothelial dysfunction. he difference in shockable rythms would support the notion of two different conditions. In addition, a Covid infection leading to a cardiac arrest probably has a high mortality in itself, meaning that patients surviving the initial resuscitation not only have to survive all usual complications but also their Covid disease.

ANSWER: We are grateful for providing these interesting explanations. We added them in the discussion.

  1. Shockable rhythm. I was surprised to see that the large difference in shockable rhythm did not result in a significant statistical difference (page 3, line 82). Doing a simple chi-square test with data extracted from your table 1 yielded a highly significant difference (see print screen below). What is the explanation for this discrepancy?

ANSWER: We supplemented the data with an article that was published in February 2021. Currently, rhythm-related relationships are as follows:

Minor comments

Should be more focused and should include a “justification” of doing this meta-analysis

ANSWER: We provided the justification in the introduction, as follows: Since CPR is a highly aerosol-generating procedure, it carries a substantial risk of viral transmission, thus discouraging the bystander resuscitation. We hypothesized that the reluctance for bystander CPR deteriorates OHCA outcomes in the COVID-19 patients. To raise the awareness to this potential problem, we performed a systematic review and meta-analysis of studies that reported OHCA in the pandemic period, comparing COVID-19 suspected or diagnosed patients vs. COVID-19 not suspected or diagnosed group.

Page 2, line 49. Please provide a reference for the 7%

ANSWER: We updated the data and changed sentence. We also added up-to-date data and literature.

Page 2, line 61. Please provide a reference for the 2.1%

ANSWER: We updated the data and changed sentence. We also added up-to-date data and literature.

Page 2, line 84. Please provide a reference for the PECOS procedure

ANSWER: Appropriate bibliographic footnote has been added.

Page 3, table 2. Do the studies included in your meta-analysis indicate, which proportion of resuscitations were performed with rescuers wearing full PPE? If yes, please include this information

ANSWER: Unfortunately, the articles taken into the meta-analysis do not include such data. We added this issue to the Limitation section.

Page 5, line 165 “lower” instead of “rower”

ANSWER: An editorial error has been corrected

Page 5, line 181. What does “consubstantial Covid-19” mean?

ANSWER: We apologise for the typo, we meant “co-existing COVID-19”.

Yours sincerely,

Aleksandra Gąsecka

Reviewer 2 Report

The paper from Borkowska and coauthors concerns the systematic revision of studies that reported out-of-hospital cardiac arrest (OHCA) during pandemic period. The authors found that the consubstantial COVID-19 in the case of OHCA reduced survival to hospital discharge rate, suggesting the improvement of cardiopulmonary resuscitation in the COVID-19 setting.

Although the analysis has several important limitations such as the limited number of studies available and the impossibility to use the influence of comorbidities, that could have affected the final elaboration, the paper is well structured and authors performed a systematic review and meta-analysis following Preferred Reporting Items for Systematic reviews and Meta-analysis (PRISMA) guidelines and with Meta-analysis of Observational Studies in Epidemiology (MOOSE) statement.

The manuscript is well written and the paper is interesting and the discussion is effective.

Author Response

Reviewer 2

The paper from Borkowska and coauthors concerns the systematic revision of studies that reported out-of-hospital cardiac arrest (OHCA) during pandemic period. The authors found that the consubstantial COVID-19 in the case of OHCA reduced survival to hospital discharge rate, suggesting the improvement of cardiopulmonary resuscitation in the COVID-19 setting.

Although the analysis has several important limitations such as the limited number of studies available and the impossibility to use the influence of comorbidities, that could have affected the final elaboration, the paper is well structured and authors performed a systematic review and meta-analysis following Preferred Reporting Items for Systematic reviews and Meta-analysis (PRISMA) guidelines and with Meta-analysis of Observational Studies in Epidemiology (MOOSE) statement.

The manuscript is well written and the paper is interesting and the discussion is effective.

ANSWER: Thank you for appreciating the value of the article and the review. 

Yours sincerely,

Aleksandra Gąsecka

Reviewer 3 Report

The authors commented on my previous concerns, however, in the meanwhile an important investigation has been published on that topic by Sultanian and colleagues in European Heart Journal (doi:10.1093/eurheartj/ehaa1067). According to my view, this investigation has to be included in the meta-analysis. Otherwise the meta-analysis won´t be up to date.

Author Response

The authors commented on my previous concerns, however, in the meanwhile an important investigation has been published on that topic by Sultanian and colleagues in European Heart Journal (doi:10.1093/eurheartj/ehaa1067). According to my view, this investigation has to be included in the meta-analysis. Otherwise the meta-analysis won´t be up to date.

ANSWER: Thank you for paying attention. This article was not available at the time of writing the manuscript. We have now included this article in the meta-analysis - as suggested - as we fully agree that this will keep the analysis up to date. Additionally, searches of databases did not reveal other articles.

Round 2

Reviewer 1 Report

The manuscript significantly improved during revision and now contains all relevant information.

Major comments:

  1. the discussion is still very lengthy and not easy to read. You should stick to the discussion of findings of your meta-analysis.
  2. the large and highly significant difference in shockable rythms could be the key explanation for the observed differences in outcome. As such, this difference should be made prominently visible in the abstract and the discusssion.
  3. Minor comment: what is the meaning of "SF" in the reporting of statistical results (line 75, line 78, etc) 

Author Response

Major comments:

  1. the discussion is still very lengthy and not easy to read. You should stick to the discussion of findings of your meta-analysis.

We updated the discussion according to the Reviewer’s recommendation.

  1. the large and highly significant difference in shockable rythms could be the key explanation for the observed differences in outcome. As such, this difference should be made prominently visible in the abstract and the discusssion.

Thank you for this suggestion. we underlined the difference in shockable rhythms in the discussion.

  1. Minor comment: what is the meaning of "SF" in the reporting of statistical results (line 75, line 78, etc) 

SF is the abbreviation from Supplementary File. This abbreviation has been explained on page 1, line 37. However, since it may be confusing, we used only the full term in the new version of the text .

Reviewer 3 Report

The authors included the investigation by Sultanian and colleagues in the revised manuscript. However, in table 1 the authors forgot to insert the patients’ characteristics of the study by Sultanian et al. Moreover, in the supplementary material the study of Sultanian and colleagues is not included in any of the Forest plots and also not in table S1, despite writing in the Results section (3.1 study characteristics) of their revised manuscript that “Detailed characteristics of the studies included in the meta-analysis are presented in the Supplementary File”. I think the authors should pay more attention and meticulously verify that all relevant data are included in tables and Forest plots before submitting the revised manuscript and supplemetary file.

The authors stated that the reluctance of bystander CPR may have led to the worse outcome of COVID patients. However, in the revised manuscript bystander CPR was not significantly different between COVID and non-COVID patients. Moreover, the study of Sultanian et al is not included in the respective Forest plot of bystander CPR in the supplementary file. So how is this statement supported?

Author Response

We apologise for this mistake. We attached the wrong supplementary file. Please find enclosed the correct file.

We also agree with the Reviewer, that the bystander CPR was not significantly different between the groups. However, we hypothesized that the quality might have been worse due to the recent data showing decreased quality of CPR in personal protective equipment. Nevertheless, since this statement is not directly supported by our data, by removed it from the discussion.

This manuscript is a resubmission of an earlier submission. The following is a list of the peer review reports and author responses from that submission.

Round 1

Reviewer 1 Report

The impact of the Covid-19 epidemic on OHCA practices and outcomes is a very interesting topic. The author comprehensively reviews and clearly summarizes the currently published reports. However, published articles for the OHCA victims with Covid-19 were only 3, which are limited and biased. Additional information is lacking in this article. If you conduct a meta-analysis after the additional results are published, you will get more important results.

Reviewer 2 Report

Borkowska and colleagues conducted a systematic review and meta-analysis of the impact of COVID-19 disease on OHCA survival rates. The results of this analysis suggested that COVID-19 disease reduced OHCA survival rates as compared to OHCA cases without COVID-19.

Globally this work is well-written. These data are of contemporary interest. However, some concerns need to be addressed:

  • A review of the studies included in the meta-analysis is not included as text in the main manuscript. The main characteristics should not be displayed only in the table 1. The authors should present in a separate section the methods (retrospective, single-center, registry etc), study population, main results and conclusion of each study. Moreover, differences among the included studies should also be presented. In my view, table 1 is rather unclear and slightly confusing.

  • There are no data on no flow- and low flow-time. I think these parameters are important prognostic parameters in OHCA. These data should be added, if possible.

  • The layout of table 1 needs to be revised. Many words do not fit entirely in one row of the tables cell. This should be corrected.

Reviewer 3 Report

I commend the authors for undertaking this meticulous meta analysis to answer a very relevant, timely and important clinical question. The pandemic has affected healthcare delivery and outcomes in many ways and has certainly affected our ability to provide care for patients experiencing out of hospital cardiac arrest. While I think this research topic would be of great interest to the reader, I do not believe that methodology of the current manuscript truly addresses the question/hypothesis set forth by the authors. Here are my reasons:

1) There are two cohorts analyzed in this study, one of which includes 'confirmed or suspected COVID-19.' Suspecting COVID-19 in a patient and having a confirmed diagnosis of it are two completely different states; especially in the middle of a pandemic, a significant number of patients we see on the frontlines, we suspect them of having COVID-19. Furthermore, our confirmation of their diagnoses is often limited by the low sensitivity of the PCR based test and also multiple co-morbidities present in these patients. Combining suspected and confirmed patients of COVID-19 leads to significant confounding in this current study and hence limits its ability to truly compare outcomes between COVID-19 vs non-COVID-19 patients. I appreciate that meta analysis limits authors' ability to tease out details between confirmed and suspected COVID-19 cases but I do strongly believe that that undertaking is a must for the findings of the current study.

2) Although not statistically significant (CI crosses 1 and P values are > 0.05), there is a clinically significant difference between the so called 'COVID-19' and 'non-COVID-19' patients in respect to several key features that affect outcomes post OHCA: 1) Shockable rhythms 2) ALS implementations 3) utilization of mechanical chest compression devices that allow for minimal interruptions in effective CPR. While no one feature met statistical significant, it is hard to believe that cumulatively they would not and furthermore, clinically they most likely would. 

3) One of the major limitations of the study is that it compares populations across three different countries. Baseline co-morbidities of each population cohort would certainly affect outcomes post OHCA and hence be a major confounder. 

Reviewer 4 Report

The authors performed a systematic review and meta-analysis of studies concerning OHCA in the CORONA pandemic period, comparing OHCA in COVID-19 suspected or diagnosed patients vs COVID-19 not suspected or diagnosed group. Data from 1,666 patients from 3 retrospective out of 240 studies were analysed, coming one each from France, Italy and Republic of Korea. The main finding – primary outcome – was survival to hospital discharge (SHD). The authors found a significantly lower SHD rate in COVID-19 than in Non-Covid-19 patients (OR 0.24) and non-significantly lower rates of ROSC and SHD with favorable neurological outcome (secondary end points). From their findings the authors concluded that their meta-analysis suggests that the suspected or diagnosed COVID-19 in the case of OHCA reduces the SHD rate.  

The following ocmments may be allowed:

In the present Corona pandemic, the topic the authors are dealing with is of utmost importance! The authors have collected the very few available retrospective registry data and analysed them as a meta-analysis in a proper way.  Of course, in fact of the low number of studies and the low methodological quality (retrospective observational data) of the studies, the meta-analysis has a high methodological bias as can be seen from the broad confidence intervals. This has been clearly stated by the authors. Nevertheless, the authors did what can be done with the available data set. Despite these limitations, the findings are of great clinical relevance, hypothesis-generating and need further scientific and clinical attention.

I have no further comments or criticism.